# Image Recovery from Synthetic Noise Artifacts in CT Scans Using Modified U-Net

**DOI:** 10.3390/s22187031

**Published:** 2022-09-16

**Authors:** Rudy Gunawan, Yvonne Tran, Jinchuan Zheng, Hung Nguyen, Rifai Chai

**Affiliations:** 1School of Science, Computing and Engineering Technologies, Swinburne University of Technology, Hawthorn, VIC 3122, Australia; 2Macquarie University Hearing (MU Hearing), Centre for Healthcare Resilience and Implementation Science, Macquarie University, Macquarie Park, NSW 2109, Australia

**Keywords:** lung cancer, noise artifacts, denoising

## Abstract

Computed Tomography (CT) is commonly used for cancer screening as it utilizes low radiation for the scan. One problem with low-dose scans is the noise artifacts associated with low photon count that can lead to a reduced success rate of cancer detection during radiologist assessment. The noise had to be removed to restore detail clarity. We propose a noise removal method using a new model Convolutional Neural Network (CNN). Even though the network training time is long, the result is better than other CNN models in quality score and visual observation. The proposed CNN model uses a stacked modified U-Net with a specific number of feature maps per layer to improve the image quality, observable on an average PSNR quality score improvement out of 174 images. The next best model has 0.54 points lower in the average score. The score difference is less than 1 point, but the image result is closer to the full-dose scan image. We used separate testing data to clarify that the model can handle different noise densities. Besides comparing the CNN configuration, we discuss the denoising quality of CNN compared to classical denoising in which the noise characteristics affect quality.

## 1. Introduction

The Global Cancer Observatory (GLOBOCAN) report estimates 19.3 million new cancer cases with almost 10 million deaths in 2020, lung cancer being the second most frequently diagnosed cancer after female breast cancer. However, lung cancer has the highest mortality at 18% [1] as the diagnosis often occurs at the late cancer stages. A randomized trial for lung cancer screening in the Netherlands and Belgium, which started in 2003, showed an increase in patient survivability as early cancer detection led to early treatment [2]. Early detection of cancer using cancer screening, targets people at high risk of lung cancer, such as heavy smokers and people with a family history of lung cancer. Screening is through CT scans, which utilize lower dose radiation while maintaining image quality just enough for the assessment [3]. The concept began in 1990 in which the image has enough detail to detect lung parenchyma but suffers image degradation from increased noise [4]. Nowadays, low radiation scans, known as Low Dose CT (LDCT), have become a popular choice for screening. These use approximately 25–30% of the standard dose radiation [5]. However, the LDCT has an increased noise level, leading to a risk of incorrect assessment. For example, it was found that cancer screening using LDCT of 53,454 people resulted in 24.2% positive, but 95% were false positive [6].

Tube current, collimation factor, bowtie filter, and exposure time are factors that can affect radiation dose during scanning. The radiation dose is proportional to the photon count, coupled with the absorption and scattering tendency during the scan, which may result in the detector receiving fewer photons. As a result, noise may appear in the projection data. The noise changes into a streak line after image reconstruction. There is also electronic noise on the image, such as shot noise, although in a modern CT scanner, the impact of electronic noise is less apparent due to a better electronic circuitry design [7].

Noise removal from low-dose CT scan images has become a growing area for research. Denoising methods have expanded from classical denoising such as Gaussian Smoothing, Non-Local Mean (NLM), to 3D Block Matching (known as BM3D) into using various CNN models, each with its own strength. Denoising CNN requires a learning process that depends on full-dose and low-dose radiation images, and it is impossible to get an exact copy of the image due to possible patient movement or even motion artifacts in between scans [8] and the risk of radiation, even though some researchers have attempted to have two runs within the scan, with a radiologist as the observer [9,10].

In classical denoising, there has been research to improve BM3D to address CT scan noise using a context-based method. The quality score was based on Contrast to Noise, which improved around 30% on average from the original BM3D [11]. There is various research on using CNN for denoising. A Generative Adversarial Network (GAN) method involved training denoising quality with a generator utilizing control of a discriminator on a 3D cardiac CT, resulting in four points of Peak Noise to Signal Ratio (PSNR) improvement [12]. There has been research using an offline dictionary (database of patches) to reconstruct low-dose CT by restoration using similar image patches. The quality measurement used radiologist scores [13,14]. Several modifications of U-Net for CT scan denoising have been used by reducing the filter size and adding an extra layers per stages [15], by removing a concatenated layer and adding a recurrent-residual connection [16], by using a stagnant filter size of 128 on all stages, and using a Leaky Rectified Linear Unit (ReLU) as the activation function [17]. There has been research using different network architecture and processes on targeting the residual noise to train in Residual Network (ResNet) [18].

As the main contribution of this research, we propose a denoising solution that stems from the understanding of several CNN models. The proposed model is a stacked modified U-Net model. The modification includes a transpose convolution layer for the decoder stage, increasing spatial learning using convolution layer stride-2 for contraction, improving noise detection using a high filter number at the early layer, maintaining the number of filters across all layers, and stacking two models together for improvement. These modifications have not been used together before to deal with the denoising problem. The dataset used synthetic noise obtainable by using a noise insertion tool at 10% of the standard dose. The approximate difference was 5–6% of the actual screening images. The structure of this paper is as follows: Section 2 covers the methodology of determining the CNN layers and the model. Section 3 describes results and comparative quality improvement before and after denoising, and comparative study of another method. Section 4 discusses the impact of denoising on a different dataset. The conclusion is provided in Section 5.

## 2. Materials and Methods

### 2.1. The Data Set

The data set used in this study was from the research of LDCT-and-Projection-data, stored in the Cancer Imaging Archive (TCIA) library. We utilized six patient datasets (out of 299 patients in the pool) with 1906 image pairs to train the neural network and test the result independently. Image pairs means each standard-dose image has a synthetic low-dose image copy. The sample dataset was the scan result of the Somatom Definition AS+ and Somatom Definition Flash CT scanner’s scan result from a patient with and without solid non-calcified nodules [19].

### 2.2. Noise Simulation

Noise simulation was obtained using a noise insertion tool that utilizes a validated photon-counting model applied to the post-processed projection data of a standard-dose scan. The images had a CT scan setting of 1.5 mm slice thickness, 120 kV, and a tube current between 650–750 mA for the full-dose scan. The projection data were extracted from the scanner to add the synthetic noise using a known noise model, then the modified projection data were returned to the scanner to utilize weighted, filtered back-projection (FBP). The simulation target was 10% of the full-dose radiation dose for the chest scan [12,20].

A theoretical approach for noise simulation, by altering projection data using Gaussian noise, was considered and tested. The projection data were calculated from a standard-dose image using Radon transformation.
(1)$$\rho \left(\omega ,\theta \right)=\int_{-\infty }^{\infty }\int_{-\infty }^{\infty }f\left(x,y\right)e^{-i2\omega \left(xcos\theta +ysin\theta \right)}dxdy$$
where $\rho \left(\omega ,\theta \right)$ is projection data (sinogram) of the CT scan image $f\left(x,y\right)$ (1). The detected photon count (attenuated intensity) was traceable using the Beer-Lambert Law for an non-homogenous material and a bowtie filter profile.
(2)$$I=I_{0}\cdot P_{b}\cdot e^{-\int \mu \left(x\right)dx}=I_{0}\cdot P_{b}\cdot e^{-\rho \left(\omega ,\theta \right)}$$
where I is the detected photon count (count per second—cps), and $I_{0}$ is the original photon count (incident flux—cps), $P_{b}$ is the bowtie profile and $\int \mu \left(x\right)dx$ is the non-homogeneous target equivalent to the projection data $\rho \left(\omega ,\theta \right)$ (2).
(3)$$I_{0}=K\cdot c\cdot mA\cdot s$$

The incident flux $I_{0}$ depends on the scanner constant K, collimation factor c, radiation dose mA, and exposure time s (3).
(4)$$I_{0L}=10\%*I_{0}$$
(5)$$I_{L}=P_{s}\left(I_{0L}\cdot P_{b}\cdot e^{-\rho \left(\omega ,\theta \right)}\right)$$

The simulation of low-dose radiation $I_{0L}$ was achievable by reducing the incident flux value to 10% of the standard dose (4). The implementation of random Poisson distribution $P_{s}$ generates noise in the new attenuated intensity $I_{L}$ (5).
(6)$$\rho_{L}\left(\omega ,\theta \right)=-ln\left(I_{L}/I_{0L}\right)$$

The synthetic low-dose projection data $\rho_{L}\left(\omega ,\theta \right)$ is recalculated using Formula (6), with inverse Radon transformation to reconstruct the image [5]. This research focuses on neural networks, and there was no testing on the actual CT scanner. Thus, it was impossible to obtain the bowtie filter profile, exposure control, and electronic noise. Furthermore, back tracing the image into the projection data can cause deviation from the projection values, leading to greater error in the denoising results.

### 2.3. Selection of Denoising Technique

There are many approaches to address the image denoising problem, from the classical process to the recent utilization of neural networks. Classical denoising aims to remove noise by calculating the pixel grey value to the spatial or transform domain. In comparison, network training in neural networks aims to distinguish noise features and remove them [21]. We considered three classical denoising methods that use a spatial or transform domain to compare the denoising quality of the proposed CNN. The three methods were Gaussian smoothing, Non-Local Mean (NLM), and 3D block matching (BM3D). Gaussian smoothing applies a Gaussian function to the image to reduce noise using a smooth blur. The NLM method was introduced in 2005 for image denoising by weighting all pixels in the image with similarity to the target pixels [22]. While the BM3D method was introduced in 2007 using the same grouping method as NLM using thresholding, followed by applying the Wiener filter [23].

The proposed network stems from use of an autoencoder combined with CNN for image denoising. Several design layers fit into solving the denoising problem, each with a minor modification, even though the original development intention was for segmentation or classification. These designs became a reference for developing the new, proposed network. The first reference was U-Net, created in 2015 to deal with image segmentation and classification problems. The first half of the model consists of encoder layers that use a dual 3 × 3 kernel for the convolution process and a Rectified Linear Unit (ReLU) activation for each layer. The encoder layer uses a Max Pooling layer for contraction to move into deeper encoder layers. The second half consists of the decoder layer with the same dual 3 × 3 kernel for the convolution process and ReLU activation. This uses transpose convolution on stride 2 for expansion and skips the concatenation link from the corresponding encoder layer. There is a dropout layer before the last pooling and the first transpose convolution layer. Initially, the network used the SoftMax layer to assign probability distribution on the classification process [24]. Changing it into a Regression layer enabled the network to deal with the denoising problem. The number of filters in U-Net start at 64 and double every encoder layer with the deepest encoder layer having 1024 filters. Some researchers have modified the layer depth. The number of features to improve the training loss with each modification depends on the noise characteristic. The denoising target varies from an image with Gaussian noise [25] and a low-dose CT image (using noise simulation through Poisson distribution on the projection data) [26].

The second reference was the Segmentation Network (SegNet), developed in 2016 for dealing with the segmentation problem. It has a similar structure to U-Net, including a dual 3 × 3 kernel for the convolution process. The design utilizes the Batch Normalization layer before the ReLU activation layer and Un-polling skip connection [27]. The third reference is derived from the Deconvolution Network (DeConvNet) developed in 2016 for semantic segmentation. DeConvNet has the same characteristic as the SegNet, except the decoder layers utilize transpose convolution [28]. The original SegNet and DeConvNet have the same filter number of 64 across all layers. We increased the filter number to 128 for comparison purposes. Replacing the Softmax layer with a Regression Layer in both and removing the Batch Normalization layer enabled the SegNet or DeConvNet to deal with the denoising problem.

The fourth reference came from Residual Encoder-Decoder Network (RedNet), developed in 2016 for image restoration. RedNet uses encoder and decoder layers without a contraction or expansion process. The activation layer is ReLU, the decoder layer uses the transpose convolution process, and the summation layer is the skip connection. Without the contraction and expansion layer, the training can reduce spatial degradation and improve memory usage [29]. RedNet has the same number of filters (128) across all layers and has been used for CT scan denoising using Poisson noise on projection data [30]. The last reference came from stacking a model together, the Stacked U-Net, which was used to set-up two or more U-Net together for segmentation in 2018 [31] and 2019 [32], and for image reconstruction in 2021 [33].

The previous stacked U-Net models were used on different applications, and more modification was required to solve the denoising problem and to optimize the computational time, such as reducing the number of U-Net in the stack and increasing the convolutional operation in each block stage. The modification make the model different from the original, and for that reason there is no comparison to the existing stacked U-Net included in this paper. We only considered the stacked setting from those models without following the stack number in the existing models.

The proposed network contains two stacked U-Net layers, each with encoder and decoder layers with 128 as the number of filters on each layer. While the comparison network uses an encoder depth of 4, our model reduced it to 3. The encoder layers use a triple 3 × 3 kernel for the convolution process and Rectified Linear Unit (ReLU) activation for each layer; an extra convolution layer on stride 2 is for contraction to the deeper encoder layer.

This approach, which does not use a pooling layer, optimizes weight and biases and improves model accuracy [34]. The decoder layers use a triple 3 × 3 kernel for the transpose convolution process and Rectified Linear Unit (ReLU) activation for each layer. The expansion utilizes another transpose convolution on stride 2. The utilization of transpose convolution for decoder layers follows the idea from DeConvNet and RedNet. The skip connection uses the concatenation layer between the corresponding encoder and decoder on each network and between the stacked network (Figure 1). The input image layer has 512 × 512 pixels without a downscaling process to maintain the original spatial pixels. Training uses an Adaptive Moment Estimation (Adam) optimizer on its default settings to update the weight. Adam performs better than Scholastic Gradient Descent (SGD), Root Mean Square Propagation (RMSProp), Adaptive Gradient (AdaGrad), and Adaptive Delta (AdaDelta) [35]. Two previous studies used the same optimization for CNN image denoising [36] and MRI image denoising using the feature extraction method [20].

Each block before contraction or expansion layer may have different convolutional layers, one and two are the most common. Figure 2 shows the naming arrangement for each layer number in a block in this paper.

### 2.4. Performance Analysis

Network training used Root Mean Square Error (RMSE) to calculate the loss (difference) between the target image and the training output.
(7)$$RMSE=\sqrt{\overset{Px}{\underset{i=1}{\sum }}\left({\left(t_{i}-o_{i}\right)}^{2}/Px\right)}$$

If $t_{i}$ is the target pixel, $o_{i}$ is the output pixel after each training epoch, and Px is the number of pixels in the image (512 × 512 = 262,144), then the *RMSE* formula is as per Equation (7). A few analysis methods are available to determine the denoising quality; Peak Signal to Noise Ratio (*PSNR*) measures the ratio between peak signal against noise in the image. This can determine the amount of noise before and after denoising related to the maximum value of the image pixel. A higher *PSNR* score indicates less noise in the image, and score improvement can show the effectiveness of the denoising process.
(8)$$PSNR=20{\;\log}_{10}\left(f_{max}/\sqrt{mse}\right)$$

*PSNR* values are on a logarithmic scale between the maximum signal ($f_{max}$) and the mean square error (mse) (8). The mean square error is the average square of the difference between the target and the reference image. The aim is to find the improvement of *PSNR* score between the low-dose CT scan images and the denoising result.
(9)$$SSIM_{\left(t,r\right)}={\left[l_{\left(t,r\right)}\right]}^{\alpha }+{\left[c_{\left(t,r\right)}\right]}^{\beta }+{\left[s_{\left(t,r\right)}\right]}^{\gamma }$$

The second method uses a Structural Similarity Index (*SSIM*) to determine the image’s luminance (brightness), contrast, and structural information compared to the reference image [37]. This method allows checking the score improvement before and after the denoising process. A higher *SSIM* score indicates a close resemblance to the reference image. *SSIM* scores depend on the calculation of three components, the luminance $l_{\left(t,r\right)}$; the contrast $c_{\left(t,r\right)}$ and the structural information $s_{\left(t,r\right)}$ (9). The quality improvement follows the same process as *PSNR* for this method.
(10)$$NIQE=\sqrt{{\left(v_{1}-v_{2}\right)}^{\tau }{\left(\Sigma_{1}+\Sigma_{2}/2\right)}^{-1}\left(v_{1}-v_{2}\right)}$$

The third method uses a Natural Image Quality Evaluator (*NIQE*), a blind image quality assessment utilizing measurable deviation from statistical regularity on the image patches [38]. The NIQE score is calculated based on the NIQE model. The model uses Natural Scene Statistic (NSS) obtainable from Generalized Gaussian Distribution (GGD) and Asynchronous GGD (AGGD) fitting of the Mean-Subtracted Contrast Normalized (MSCN) image collection [39]. A lower NIQE score indicates a better-quality image. The calculation is based on the vector and covariance of the NIQE model ($v_{1}$ and $\Sigma_{1}$) and the target image ($v_{2}$ and $\Sigma_{2}$), with the sharpness threshold of the patch T (10). The patch size is 48 × 48 and the sharpness threshold is 0.5. Because this method uses statistical modeling, the first approach to find image quality improvement is to obtain the score for low-dose images (noisy references).
(11)$$NIQE_{I}=NIQE_{LD}-NIQE_{D}$$

As the scores become lower with better image quality, the improvement calculation ($NIQE_{I}$) is obtained from the difference between the noisy reference NIQE score ($NIQE_{LD}$) and the denoising result score ($NIQE_{D}$) (11).

## 3. Results

### 3.1. Noise Synthesis

The conversion from full-dose and low-dose of Digital Imaging and Communications in Medicine (DICOM) images into a 512 × 512 array was done using MATLAB software. An attempt to create synthetic noise from the formula did not provide good results because of the lack of Bowtie profile information. Utilizing the noise model and Bowtie profile provides clearer streak lines focusing on the area of interest with maximum width as seen in the sample images.

Without the model and Bowtie profile, the noise had fewer lines with a broader noise effect (Figure 3). This Bowtie effect created focused noise on the target area, which affects the selection of denoising methods.

### 3.2. Observed Network Setting

The dataset was split into two parts. The first consisted of three patients with 986 image pairs, the training utilized a random 50% of the image pairs (493 images of low-dose and 493 images of standard-dose), validation used 30% of the image pairs, and testing used 20% of the image pairs. Training used a supercomputer with a 64 GB GPU and 80 GB CPU memory and MATLAB 2021b software. The performance scores were derived from the training time of each CNN model, including the reference models for comparison. The MATLAB version was used since it provides a new function for capturing the lowest RMSE validation loss, which negates the need to use training early stopping.

The network used the ADAM optimizer setting with default parameters of β1 = 0.9, β2 = 0.999, and ε = 10^−8^ [31]. There were 300 training epochs in a small batch of two images. Having the entire image of 512 × 512 as an input utilized a lot of memory, and it was impossible to train with a larger batch. Training used a learning rate of 10^−4^, with a reduction of 0.8 every ten epochs. At 300 epochs, the final learning rate was 1.5 × 10^−7^.

The proposed model used two stacked networks, considering training improvement and efficiency of training time. The selection stacked network over a single network reduced the RMSE loss by 310 points and doubled the training time from 544 to 1229 min using the same 300 epochs. Considering time efficiency, when the single network stopped the training at the same 544 min, the stacked network still had a lower RSME loss by 237 points (Figure 4).

Filter size determines the observation area. A smaller filter of single 3 × 3 can run fast but has limited coverage. Tests on larger size 5 × 5 and 7 × 7 filters produced excellent RMSE losses, at the cost of long training times. Since a dual 3 × 3 filter has the same coverage as a single 5 × 5, while a triple 3 × 3 filter can cover a single 7 × 7 at a reduced number of weight parameters, both were tested to check the quality and training time of the proposed model.

The training result showed an increase in the training time from 816 min for a single 3 × 3 kernel to 1229 min for a double 3 × 3 kernel and 1564 min for a triple 3 × 3 kernel. There was an improvement in RMSE loss when using more filter layers in one encoder/decoder stage, indicating layer saturation. The RMSE loss improved by 363 points on a double 3 × 3 and improved by 60 points on a triple 3 × 3 (Figure 5).

At the same time, this shows that a dual 3 × 3 kernel had the same result as a single 5 × 5 kernel with almost half of the training time of 2191 min. At the bigger kernel size of 5 × 5, the learning rate no longer fit, as the training result showed higher oscillation up to the 70 epoch (500 min) (Figure 6).

The proposed model used a dual 3 × 3 instead of a triple 3 × 3 or a single 5 × 5/7 × 7 kernel due to the slight increase in PSNR and SSIM improvement shown in the result but utilized more memory and training time (Table 1). AV status in the header shows the average score of 174 testing images, each representing quality components NIQE, PSNR, and SSIM. The green highlight indicates the proposed network setting in quality comparison to different network setting.

The final piece of the network setting for testing was the number of filters. The number of filters was set at 128 for all layers as this provides the most optimum setting for training. The doubling up method on the filter numbers, as seen in U-Net, did not provide an improved result. Crux lies in the noise focus needed a high filter number in the early layer. The 64 filters provided a short training time (510 min) but with a higher RMSE loss of 895 points compared to the 128 filters. When the filter was increased to 256 it had a worse RMSE loss by 179 points while increasing the training time substantially (3013 min) (Figure 7).

### 3.3. Comparison between Denoising Techniques

Three classical denoising methods (Gaussian smoothing, NLM, and BM3D) were tested on the same image samples for quality comparison of the CNN methods, and the quality between CNN models. The proposed network model has a label of “S-Dual3” in the following bar charts, indicating the use of a dual 3 × 3 layer at each encoder/decoder stage. The other CNN models are a RedNet with 20 layers, a modified U-Net, a modified SegNet, and a modified DeConvNet. These networks retained the model structure with modifications, as stated earlier.

Table 2 indicates the training time difference between CNN models with the proposed network run for the longest time. The figures does not include classical denoising as they do not need any training.

A comparison chart showed average scores on three different quality measurements, NIQE, PSNR, and SSIM. The mean score was derived by averaging the quality score of 197 images. The NIQE comparison chart (Figure 8) shows the score of low-dose images as the X-axis, which are presented as a group to understand the chart better. The X-axis scores were obtained by measuring the low dose image against the full-dose image quality modeling result. As the chart does not accommodate the number of images in one group, there is a discrepancy between the improvement score and the total mean score of each denoising method. The Y-axis represents the NIQE improvement score which is the amount of reduction from the NIQE score of the low-dose image. There is a cap on the lower end of low-dose image NIQE score at 16; any score at or below 16 causes the denoising process to exhibit a higher score. A higher score indicates a worsening quality. Tracing back to the comparison score between low-dose ($NIQE_{LD}$) and full-dose image ($NIQE_{FD}$) shows that the latter has a higher score than the former ($NIQE_{LD}$ < $NIQE_{FD}$).

The bar chart is configured with average NIQE scores from low (left) to high (right). The average score was calculated over all tested images, unlike in Figure 8 which shows classes of noise level. The S-Dual3 method was best with 14.89 points of improvement, followed by SegNet (14.80), DeConvNet (14.73), UNet (14.25), RedNet (14.00), BM3D (13.60), NLM (12.42), and Gaussian (11.47).

The PSNR comparison chart (Figure 9) shows the PSNR score of low-dose images on the X-axis, while the Y-axis represents the PSNR improvement score. The X-axis grouping and bar arrangement follows the same configuration of low (left) to high (right) as the NIQE chart. The S-Dual3 method was best with 10.55 points of improvement, followed by U-Net (10.01), RedNet (9.85), DeConvNet (9.72), SegNet (9.67), BM3D (6.33), Gauss (4.77), and NLM (2.28).

The SSIM comparison chart (Figure 10) has the same presentation as the previous two charts. The S-Dual3 method was best with 0.00895 points of improvement, followed with RedNet (0.00873), DeConvNet (0.00872), U-Net (0.0087), SegNet (0.0087), BM3D (0.00731), Gauss (0.00452), and NLM (0.00365).

Both PSNR and SSIM charts show a similar equalizing trend with a higher denoising effect on higher noise density. There was a slight difference in these two quality scores between CNN models, with the proposed model showing the highest score and observable difference. The classical denoising techniques had lesser scores with significant discrepancies than the CNN techniques. This result indicates the inability of classical denoising to address CT scan noise that occurs with different densities (focused noise).

## 4. Discussion

This study compared the visual observation of a denoising image against a quality score to obtain a relationship between those two. A suitable location would be either the upper thorax or diaphragm area, in which the area of interest contains a rich contrast variation.

Figure 11 shows the full-dose image and its counterpart of the synthetic low-dose image from the upper thorax area from one of the data samples. The number indicates a PSNR score of 50.7883, with only one quality score for comparison. A noise mask was created by taking the difference between those two images and applying the contrast setting at 10% of the maximum pixel value. The noise mask shows a higher noise density in the area of interest (patient’s body) that appears as a short line compared to the other area, which resembles “shot” noise. The denoising result from various techniques shows image quality with an accompanying noise mask to better understand the amount of noise that has been removed. The denoising noise mask is derived from the difference between the result and full-dose image, applied the same contrast setting as the previous mask to retain the same reference (Figure 12).

Classical denoising had less impact on removing noise. The NLM technique reduced some noise outside the area of interest and could not deal with focused noise, with a slight increase of PSNR score less than 1 point. Gaussian smoothing reduced a lot of noise; thus, the high PSNR score improvement at 6.7112. The downside was the introduction of blur to the image, which had a detrimental effect on assessment. The BM3D technique provided better noise removal than NLM, even reducing the impact of focused noise, although the limitation appeared at higher noise density, with a PSNR score improvement at 2.9458. The SegNet and DeConvNet techniques had close PSNR improvement scores of 10.6364 and 10.7092; the denoising result and noise mask were similar. Both models had differences in the decoder stage between a standard convolution layer and transpose convolution layer. The slight improvement indicated that the application of transpose convolution into the decoder stage gave a better result.

The RedNet technique was slightly worse than SegNet, DeConvNet, U-Net, and the proposed model, with a PSNR improvement score of 10.3974. The difference was the missing contraction + expansion layer in the model, creating lesser quality. The U-Net had the second highest PSNR improvement score at 10.8335. The noise mask showed less noise, but the image showed higher contrast than the full-dose image. This contrast is because of the Max Pooling layer. The same effect appeared using the RedNet technique. The proposed model had an improvement score of 11.5187, and showed slightly more noise in the mask than U-Net, but resembled the full-dose image.

Two additional image set (Figure 13 and Figure 14) were added to confirm better improvement resulting from the proposed network.

During repeated training, there is a chance that the 30% validation data set (295 images) may lean into a group of images with lower or higher noise density. The same can happen on the remaining 20% testing data set (197 images), making the NIQE, PSNR, and SSIM scores vary during each test. Furthermore, the training dataset had different areas of interest due to the patient size, creating a difference in noise density. Confirming the CNN network can address CT scan images in general, so a completely new dataset was used for comparison to guarantee the best denoising result. The second part of the dataset was used for independent testing, and consisted of three patients with 920 image pairs.

The testing phase used only 50% of the synthetic low-dose image in the pool (460 images), and their standard dose pairs were used for quality score calculation. We used a different CT scan image set to confirm the network quality was still working well with different sizes in the area of interest. This comparison targeted only on CNN models while excluding the classical denoising method. The NIQE, PSNR, and SSIM comparison charts (Figure 15, Figure 16 and Figure 17) show a similar result. NIQE scores were worse when dealing with noise below 16 on the X-axis. The PSNR chart shows a surprising improvement for the proposed model with a greater PSNR score (X-axis—less noise), better than the other models. The proposed model led the improvement score for SSIM as well. The equalizing trends are visible on both PSNR and SSIM charts; more noisy images improved than less noisy ones.

This training result was tested against the actual screening image showing each method’s denoising quality. The screening dataset was obtained from National Lung Screening Trial research [40], stored in the Cancer Imaging Archive (TCIA) library [41]. We run the test using seven methods, since the BM3D requires a good image to calculate the standard deviation. Figure 18 shows that the NLM method left a trace of noise, the Gaussian caused a little image blur, and the SegNet, and DeConvNet shared the same spotty pixels i the solid area upper left and right and had contrast enhancement. The last three images had similar results to the RedNet, and UNet had higher contrast enhancement than the S-Dual3.

## 5. Conclusions

The proposed CNN model was developed to denoise low-dose CT scan images and compare various CNN models. This research proved that using particular layers and having specific models can improve the denoising outcome. The denoising result showed a close resemblance to the standard-dose image compared to other models. It illustrated that quality scores could guide quality improvement, but could not determine the same perceptual quality.

The proposed model has a suitable depth and optimum setting considering memory usage, training time, and denoising quality. A possible direction for future improvement is utilizing different layer combinations and depth arrangements. Because of resemblance of results to the standard-dose image, the model is suitable for segmentation and classification work. Caution is recommended because of the slight noise variation from the low-dose image during training, and deviation of the denoising result from the standard-dose image.

## Figures and Tables

**Figure 1 sensors-22-07031-f001:**
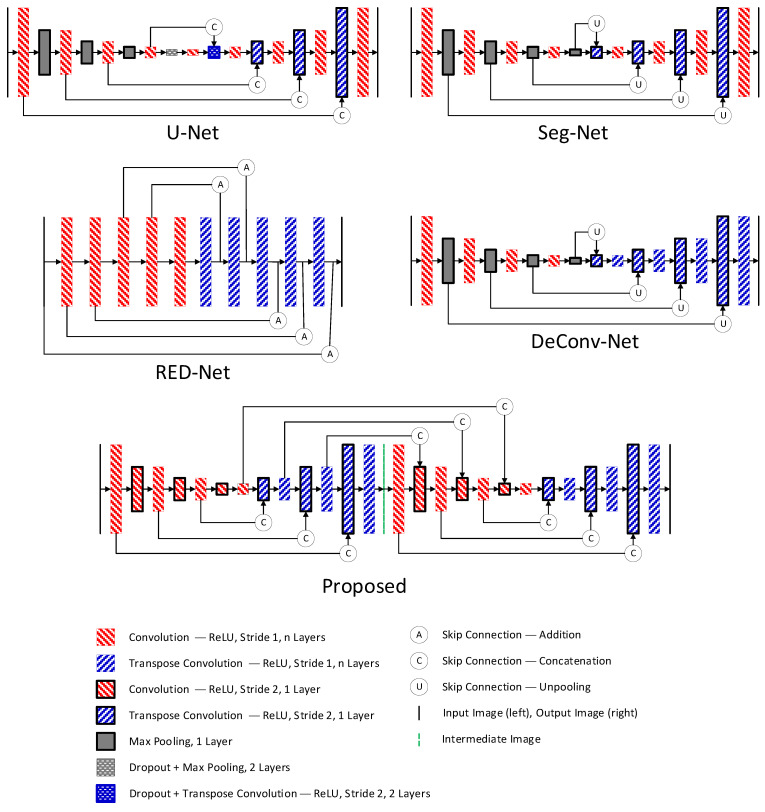
Comparison diagram between the model reference and the proposed model, including a slight modification to fit the denoising problem. The convolution and transpose convolution stride one blocks can have more than one layer (*n* Layers) (i.e., single layer with 3 × 3 filter, dual layers with 3 × 3 filter).

**Figure 2 sensors-22-07031-f002:**
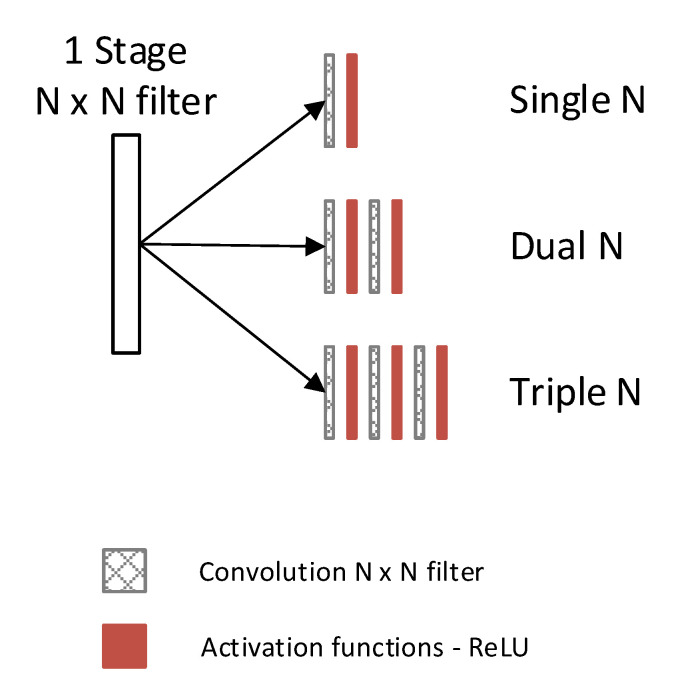
Each block in Figure 1 can become one of the options during testing. The naming in this paper is defined as Single N, Dual N, and Triple N.

**Figure 3 sensors-22-07031-f003:**
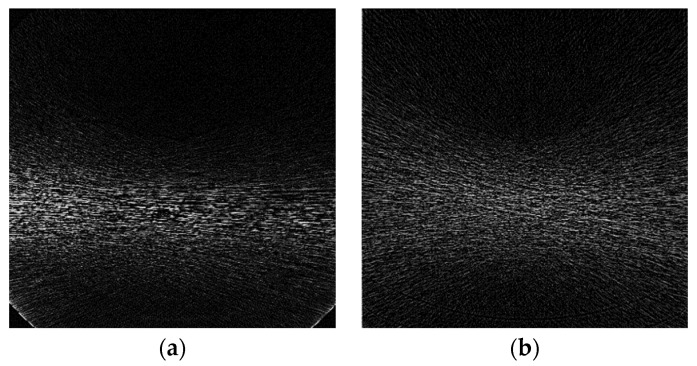
Synthetic noise mask from the sample (**a**) with the Bowtie filter; (**b**) without the Bowtie filter.

**Figure 4 sensors-22-07031-f004:**
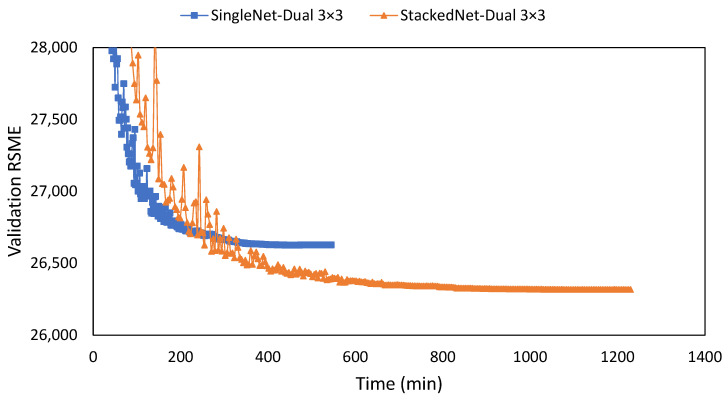
Training result between a normal-modified U-Net and the double stacked-modified U-Net on the same double layers with 3 × 3 filters. Rather than the epoch number, the X-axis represents time in minutes to show the training length for 300 epochs.

**Figure 5 sensors-22-07031-f005:**
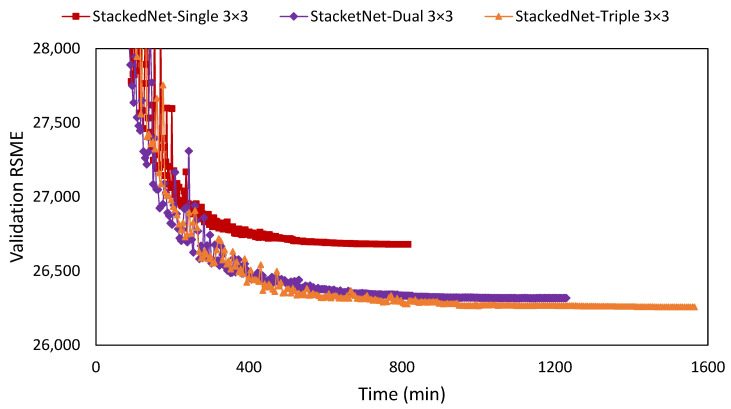
Training results of the double stacked-modified U-Net on different layers: single, dual, and triple layers with 3 × 3 filters. Rather than the epoch number, the X-axis represents time in minutes to indicate the training length for 300 epochs.

**Figure 6 sensors-22-07031-f006:**
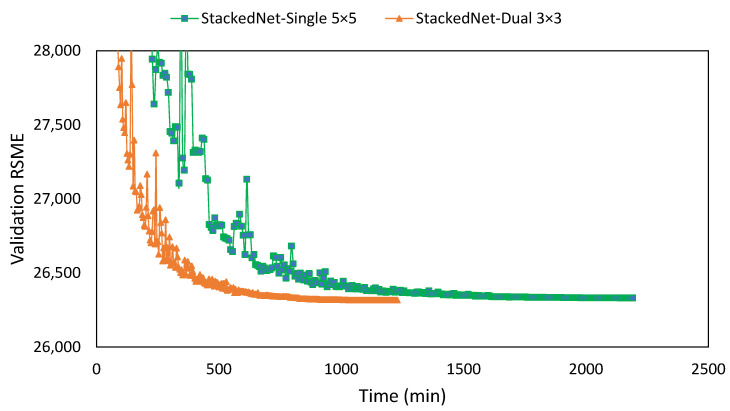
Training results of the double stacked-modified U-Net compare a dual layer with 3 × 3 filters and a single layer with 5 × 5 filters. Rather than the epoch number, the X-axis represents time in minutes to show the training length for 300 epochs.

**Figure 7 sensors-22-07031-f007:**
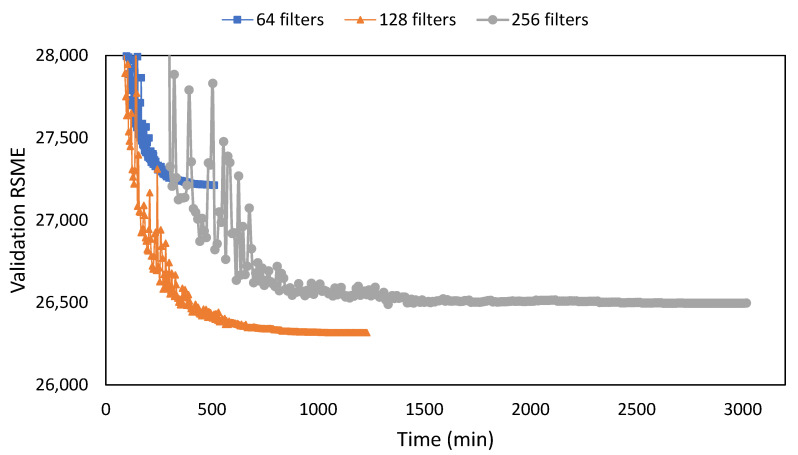
Training results between different filter numbers for 300 epochs with the X-axis representing training time.

**Figure 8 sensors-22-07031-f008:**
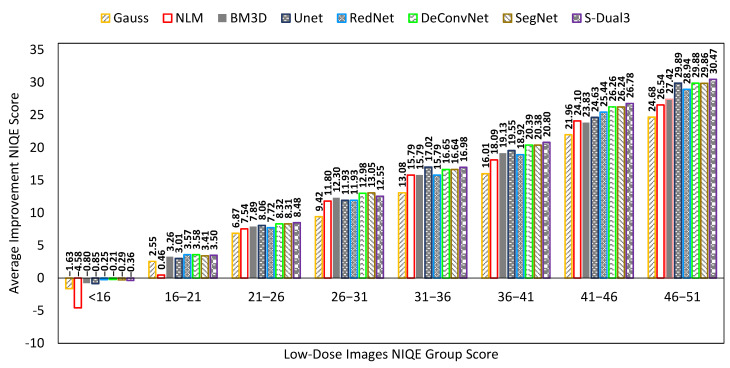
Comparison chart on NIQE scores between denoising methods. The X-axis represents the NIQE scores of the low-dose images in groups, while the Y-axis represents the amount of NIQE score improvement. Greater improvement indicates a better image quality.

**Figure 9 sensors-22-07031-f009:**
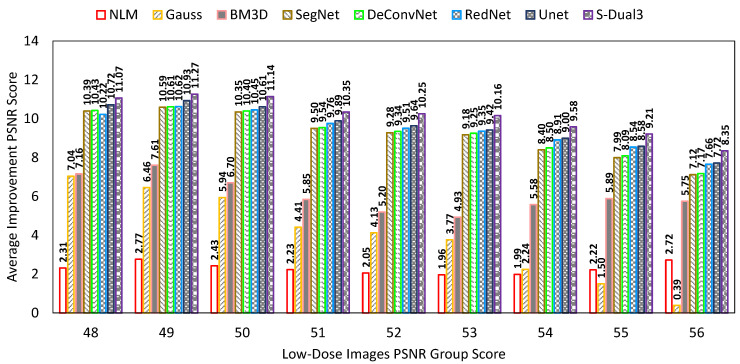
Comparison chart on PSNR scores between denoising methods. The X-axis represents the PSNR scores of the low-dose images in groups, while the Y-axis represents the amount of PSNR score improvement. Greater improvement indicates a better image quality.

**Figure 10 sensors-22-07031-f010:**
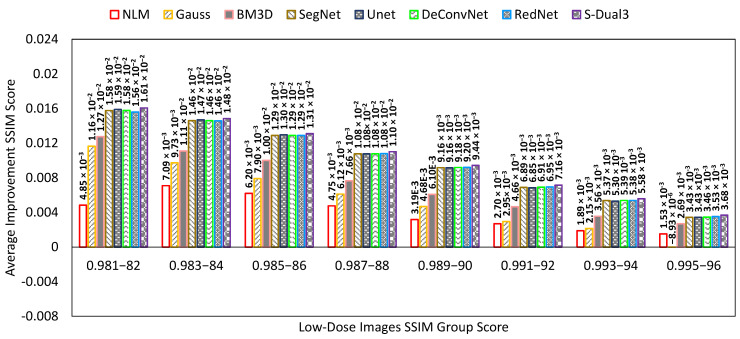
Comparison chart on SSIM scores between denoising methods. The X-axis represents the SSIM scores of the low-dose images in groups, while the Y-axis represents the amount of PSNR score improvement. Greater improvement indicates a better image quality.

**Figure 11 sensors-22-07031-f011:**
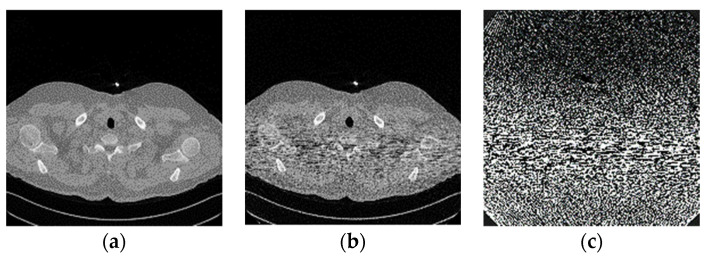
(**a**) Full dose scan image, (**b**) synthetic low-dose image at PSNR 50.7883 dB, and (**c**) noise mask for the low-dose image at 10% of the maximum pixel value.

**Figure 12 sensors-22-07031-f012:**
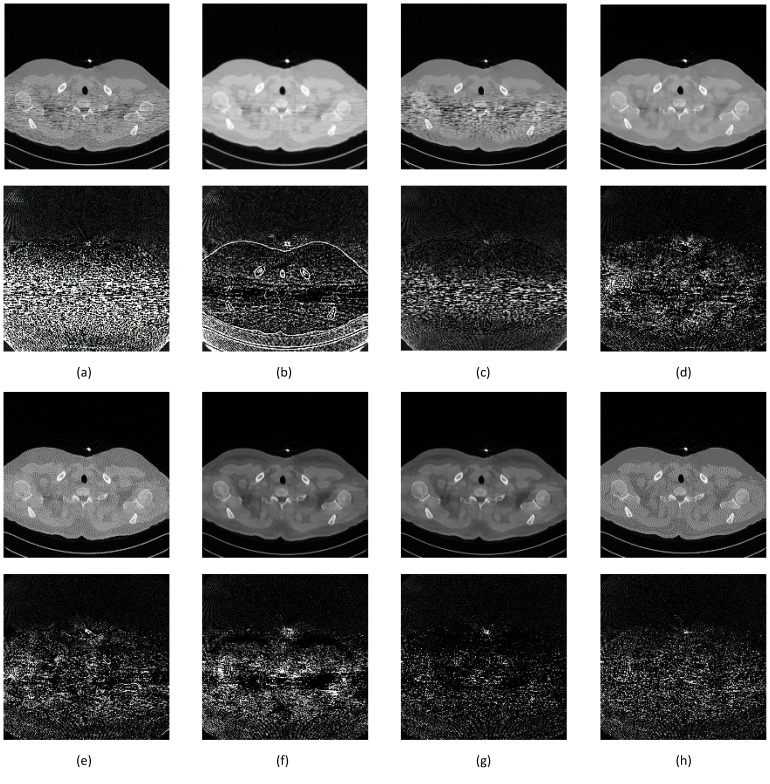
Image comparison between denoising methods on the upper thorax. The first row is the denoising result while the second row is the noise mask. (**a**) NLM, 51.3995 dB; (**b**) Gaussian, 57.4996 dB; (**c**) BM3D, 53.7341 dB; (**d**) SegNet, 61.4247 dB; (**e**) DeConvNet, 61.4975 dB; (**f**) RedNet, 61.1857 dB; (**g**) U-Net, 61.6218 dB; and (**h**) S-Dual 3, 62.3070 dB.

**Figure 13 sensors-22-07031-f013:**
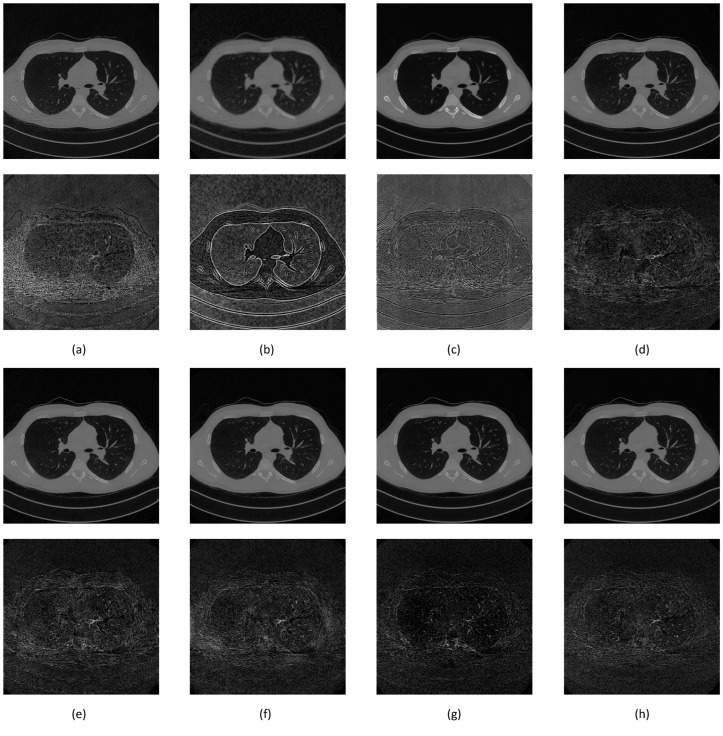
Image comparison between denoising methods in the middle lung area. The first row is the denoising result while the second row is the noise mask. (**a**) NLM, 59.7933 dB; (**b**) Gaussian, 55.5066 dB; (**c**) BM3D, 59.8350 dB; (**d**) SegNet, 61.3051 dB; (**e**) DeConvNet, 61.3086 dB; (**f**) RedNet, 61.6290 dB; (**g**) U-Net, 61.6312 dB; and (**h**) S-Dual 3, 62.0255 dB.

**Figure 14 sensors-22-07031-f014:**
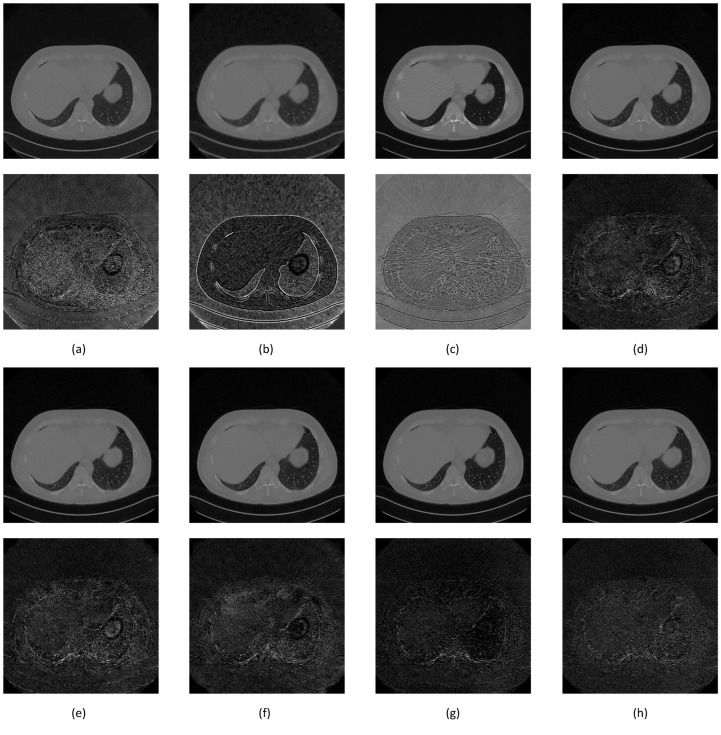
Image comparison between denoising methods around the diaphragm area. The first row is the denoising result while the second row is the noise mask. (**a**) NLM, 54.2858 dB; (**b**) Gaussian, 56.2689 dB; (**c**) BM3D, 58.3729 dB; (**d**) SegNet, 61.1182 dB; (**e**) DeConvNet, 61.1210 dB; (**f**) RedNet, 61.3288 dB; (**g**) U-Net, 61.2637 dB; and (**h**) S-Dual 3, 61.8016 dB.

**Figure 15 sensors-22-07031-f015:**
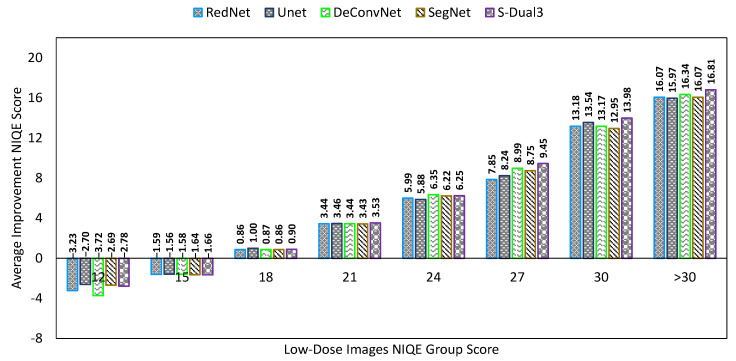
Comparison chart of NIQE scores from different CNN models with a new data set.

**Figure 16 sensors-22-07031-f016:**
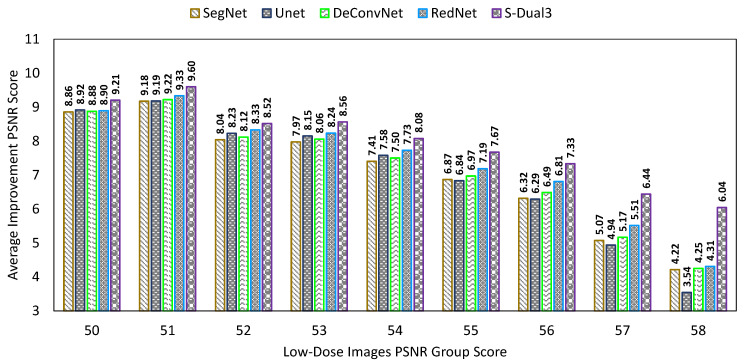
Comparison chart of PSNR scores from different CNN models with a new data set.

**Figure 17 sensors-22-07031-f017:**
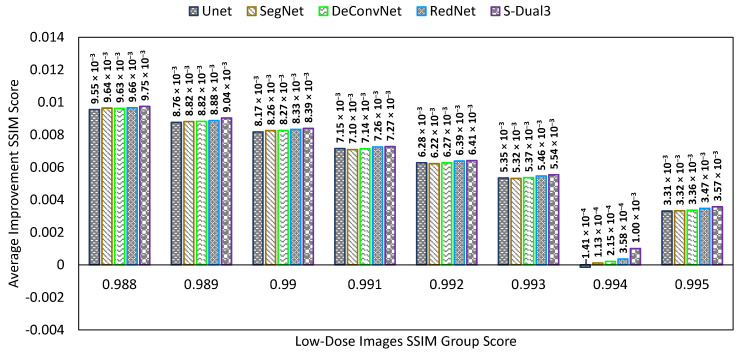
Comparison chart of SSIM scores from different CNN models with a new data set.

**Figure 18 sensors-22-07031-f018:**
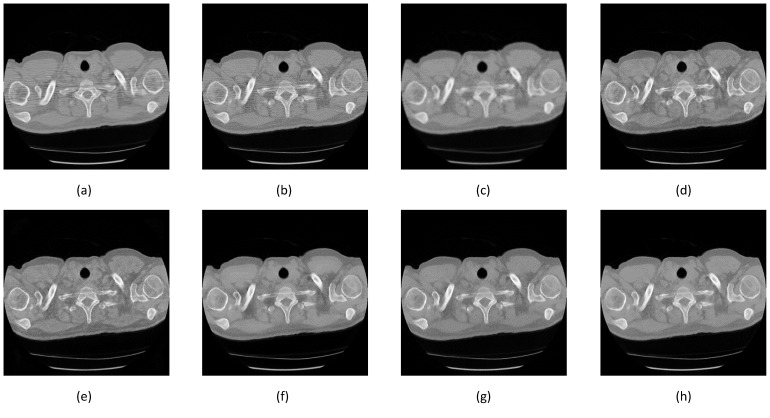
Image comparison between the screening image and after denoising. (**a**) Screening; (**b**) NLM; (**c**) Gaussian; (**d**) SegNet; (**e**) DeConvNet; (**f**) RedNet; (**g**) U-Net; and (**h**) S-Dual 3.

**Table 1 sensors-22-07031-t001:** Quality scores between various network setting on 300 training epochs.

Network	Filter	NIQE_AV_	PSNR_AV_	SSIM_AV_	Training Time (min)
Single	2 × [3 × 3]	7.672	10.377	0.00890	544
Stacked	1 × [3 × 3]	7.695	10.432	0.00892	816
Stacked	2 × [3 × 3]	7.667	10.557	0.00895	1229
Stacked	3 × [3 × 3]	7.727	10.570	0.00895	2191
Stacked	1 × [5 × 5]	7.886	10.549	0.00895	1564
Stacked	1 × [7 × 7]	7.928	10.582	0.00896	3528

**Table 2 sensors-22-07031-t002:** Training time (in minutes) comparison between CNN models at 300 epochs.

	S-Dual [3 × 3]	SegNet	DeConvNet	RedNet	UNet
**Training**	1229	644	576	764	406

## Data Availability

Publicly available datasets were analyzed in this study. Data were obtained from The Cancer Imaging Archive (TCIA) hosted by The National Cancer Institute (NCI), Washington University in St. Louis, and the University of Arkansas for Medical Sciences (UAMS), Low Dose CT Image and Projection Data (LDCT-and-Projection-data) (Version 4) [Data set] at https://doi.org/10.7937/9NPB-2637, accessed on 21 November 2021. Data from the National Lung Screening Trial (NLST) [Data set] at https://doi.org/10.7937/TCIA.HMQ8-J677, accessed on 21 November 2021.

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
