# Peer review of "Image Recovery from Synthetic Noise Artifacts in CT Scans Using Modified U-Net"

_sensors, 2022, doi:10.3390/s22187031_

Round 1

Reviewer 1 Report

Summary:

The authors proposed to modify UNet with dual-stacked and hyper-parameters tuning to solve the image denoising problem in CT Scans Image. The extensive experiments with a small dataset show the effectiveness of the proposed method against the traditional and deep learning-based denoising models.

Comments:

1. In the Abstract, the proposed method is too vague. Please explain your exact method succinctly.

2. There are so some terminologies mentioned before it is defined.

3. Why only six patients from 299? The dataset (variation) is too small for standard deep learning research. 

4. Caption of Fig. 1 seems to be an error.

5. The author should discuss the differences between the proposed method and the existing Stacked UNet.

6. The author should also compare the proposed method's performance with the Stacked UNet model.

7. The proposed model seems too complex. Even if the proposed method claimed a better performance with sacrificing time, please compare and discuss the computation time with other denoising models. 

8. Why was NIQE scores improvement of Dual3 inferior compared with existing methods?

9. For clarity, please add the exact value on the top of the bar of figs. 8, 9, 10, 13, 14, and 15.

Reviewer 2 Report

Transferring low-dose images to high-dose images through numerical methods has high clinical significance. This research is contributing to this topic. The authors of this research proposed a model for CT image denoising by stacking two networks. They justified the usage of a dual 3×3 kernel and 128 filters through numerical simulations. They compared the proposed network with other denoising networks by three image quality metrics: NIQE, PSNR, and SSIM. Based on these comparisons they validated the advantage of the proposed method.

The authors can further polish this work from the following two perspectives:

1. Please apply the proposed method to a few more CT images with different features. Using only one image is not convincing enough.

2. Please try to find open-source datasets for low-dose CT and apply the proposed method to the real low-dose images and test its performance. Typically, numerical simulation cannot cover all aspects of an imaging system. It is acceptable to generate training datasets through numerical simulations due to limit access to real data. However, in the final test step, adding some real data is preferred.

Round 2

Reviewer 1 Report

All of my concerns have been well addressed and the manuscript seems much improved.

However, there is some text that surpasses the box, such as in Figs. 8, 9, 10, 15, 16, and 17. 
